



# Beyond two water worlds: dynamic transpiration sourcing in a mixed-species boreal forest

John D. Marshall[1,2,3], José Gutierrez-Lopez[1], Daniel Metcalfe[4], Natalia Kozii[1], Hjalmar Laudon, H.[1]

[1]Department of Forest Ecology and Management, Swedish University of Agricultural Sciences, Umeå 90183 Sweden
[2]Global Change Research Institute CAS, Bĕlidla 986/4a, Brno 603 00 Czechia
[3]Department of Earth Sciences, Gothenburg University, Gothenburg 40530 Sweden
[4]Department of Ecology and Environmental Science, Umeå University, Umeå 90187 Sweden

*Correspondence to*: John D. Marshall (johnm678@gmail.com)

**Abstract.**

The water budget of a forested catchment comprises inflows, storage pools, and outflows, each with a specific stable isotopic signature. The isotopic signature is often set by summer vs. winter precipitation events, where the subsequent use of these water sources can then be traced. Streamflow, one of the major outflows, is critically important for water supplies and flooding risk. Transpiration, a second major outflow, describes the evaporation of water vapor from within tree leaves; it reduces
streamflow and is mechanistically associated with biomass production. The water sources used for streamflow and transpiration can be so isotopically distinct that they have been considered to represent two "water worlds," with distinct controls and perhaps little mixing between them. Here we describe, on a daily time-step, the contributions of water sources used for transpiration between two species that commonly occur across the Eurasian boreal zone and have close relatives in North America. Norway spruce, which is shallow-rooted, was compared to Scots pine, which roots more deeply, and both
were compared to stream water. We made these measurements in 2017, a typical summer, and compared them to 2018, a year of historic drought. Pine and spruce used distinctly different water sources. After the drought ended in 2018, the spruce switched to exclusively recent summer rainfall. Pine also switched sources, but less completely, consistent with its deeper root distributions. Streamwater was derived from residual water, with a greater representation of winter precipitation. These results support the notion that transpiration and streamwater are derived from different sources, while further dividing the transpiration
between spruce and pine. They suggest modified predictions of streamflow and forest production, especially in response to extreme weather events. Models of boreal forest transpiration should be tested against these observations to determine how well they describe this water-source differentiation.





## 1 Introduction

Water is critically needed both by vegetation (green water) and by humans, who rely on streamwater, lakes, and aquifers (blue water) (Penna *et al.*, 2018). These uses compete in the upper soil, the "critical zone" (Brooks *et al.*, 2015). Stable isotopic composition of water has shown that water moves along multiple flowpaths using different sources. Rapid improvement in stable isotopic methods presents ever-greater opportunities for delineating these processes (Penna et al., 2018), potentially providing better predictions and more effective management of both green and blue water resources, even under changing
climatic and social conditions.

Brooks et al.  (2010) highlighted the discrepancy in water sources between transpiration and streamflow in 2010, which led to the general formulation of the "two water worlds" hypothesis (McDonnell, 2014). The hypothesis states simply that 'vegetation and streams appear to return different pools of water to the hydrosphere', but the original focus was on stable isotope differences in the water sources (McDonnell, 2014). Specifically, transpired water is often enriched in the heavier $^2$H and $^{18}$O
isotopes relative to precipitation, but this is less true of streamflow. This isotopic discrepancy has been attributed to three separate causes. First, evaporation rates are higher in the surface soil than at depth, which tends to increase $\delta^{18}$O relative to $\delta^2$H in residual soil water  (Allison *et al.*, 1983). Second, the water held most tightly in small soil pores might differ isotopically from that in larger pores (Brooks et al., 2010; Sprenger et al., 2019). Third, in temperate climates, transpiration occurs mainly
during the summer, when soils may hold summer rainwater that is isotopically enriched relative to the annual mean (Allen *et al.*, 2019b).

The simple hypothesis of 'different pools of water' could be tested with single isotopes if there were strong seasonal differences in isotopic composition of precipitation. Here we test this hypothesis with continuous, *in situ* isotopic measurements of water
in transpiration and streamflow to address the role of species and seasonal differences on the partitioning of streamflow vs. transpiration.

We further tested the hypothesis that three water pools might better describe the mixed spruce-pine forests that dominate the European boreal zone. This is because Norway spruce is more shallow-rooted and hence depends almost exclusively on
summer rains for transpiration (Bishop & Dambrine, 1995; Fan *et al.*, 2017; Brinkmann *et al.*, 2019; Kahmen *et al.*, 2021). By comparison, Scots pine is often more deeply rooted than spruce (Bishop & Dambrine, 1995).  Such distinctions in water sources have been observed before, but they have been discussed primarily in terms of seasonality, e.g., summer vs. winter precipitation, rather than water "worlds" (Brinkmann *et al.*, 2018; Sprenger *et al.*, 2019; Allen *et al.*, 2019a; Floriancic *et al.*, 2024). Previous studies in Swiss forests found that spruce took up more summer precipitation than co-occurring  oak and
beech, consistent with its reputation for shallow rooting (Allen *et al.*, 2019a; Goldsmith *et al.*, 2022; Floriancic *et al.*, 2024). However, the proportion of summer precipitation in the spruces was surprisingly low.



This study quantified the existence of multiple water worlds in a boreal forest with continuous, *in-situ* monitoring of the $\delta^{18}O$ signature and seasonal origin of the transpiration stream in xylem water in spruce and pine, tree species with contrasting water-use strategies. We compared these measurements to streamwater in an average hydrological summer (2017) and in the driest summer (2018) in perhaps a century (Gutierrez Lopez et al., 2021).

## 2 Materials and Methods

This study was conducted in a 12-ha mature boreal forest catchment located inside the Krycklan Catchment study, a 67 $km^2$ catchment located in northern Sweden (64.256°N, 19.775°E) (Kozii et al., 2020; Laudon et al., 2021). The 30-year (1981-2020) mean annual temperature at the site is 2.4°C and the mean precipitation is 619 mm, of which 40% falls as snow from early November and lasting until late April. The soils in our study site were formed primarily in glacial till (84%); they commonly have an organic layer 8 mm thick (Laudon et al., 2021).

The site has a continuous forest cover, with most trees established in the early 1900's. Norway Spruce (*Picea abies)* trees dominate (61% of basal area), followed by Scots Pine (*Pinus sylvestris*; 34%) and Birch (*Betula* spp; 5%; Laudon et al., 2013). Average tree height is 23 m and the maximum is 30 m (Klosterhalfen *et al.*, 2023) The understory is typical of mature boreal forests in Scandinavia, with a dense ground vegetation dominated by bilberry (*Vaccinium myrtillus*) and lingonberry (*Vaccinium vitis-idaea*) growing over a dense moss layer (*Pleurozium schreberi* and *Hylocomium splendens*).

We compared two years with different weather conditions. The summer of 2017 was typical, but the summer of 2018 included a historically severe drought with the highest temperatures ever recorded (Lopez et al., 2021). The drought lasted from snowmelt until the end of July. Consequently, the summer of 2018 displayed the lowest summer stream runoff ever recorded at the site during almost 40 years of monitoring (Gomez-Gener et al., 2020). The drought was broken by a series of late-July rain events that provided an opportunity to observe how the stand and catchment reacted to the new precipitation pulse. Volumetric water content was measured with a vertical series of ML3 ThetaProbe Soil Moisture Sensors (Delta-T Devices, Cambridge, UK) at 5, 15, 30, and 50 cm depth.

Isotopic composition of transpiration was estimated using borehole equilibration (Marshall *et al.*, 2020). This relatively new technique has been shown to provide high temporal resolution description of water sources at natural abundance (Landgraf *et al.*, 2021), even under isotopic enrichment (Kühnhammer *et al.*, 2022). The method is a simplification of the equilibration method introduced by Volkmann et al. (2016) and later elaborated by Seeger and Weiler (2021).To collect this data, we selected 5 pine individuals (9.7 - 25.8 cm diameter, 10-22 m height) and 5 spruce individuals (6.8 to 27.5 cm diameter, 6-22 m height) that were approximately 500 m upslope from the weir where streamwater was collected and streamflow was monitored.

We drilled 15 mm diameter boreholes through the centers of the stems and connected separate PerFlouroAlkoxy (PFA) lines (Saniflex AB, Lidingö, Sweden) using a 10mm Swagelok connector that was screwed into the tree. Water vapour inside the



borehole was transported through their individual lines to a custom-built 16-valve manifold block consisting of electrically
actuated solenoid values. All lines were heated using heating cable (Nexans Sweden AB, Grimsoe, Sweden) and wrapped in
insulation to prevent condensation inside the tubing. The manifold cycled through all the trees several times per day. For this
analysis, these values were averaged into daily means for each tree. The borehole method assumes that the water vapor in the
borehole reaches equilibrium with the xylem water before it leaves the stem. Model runs and data suggested that the water
vapour in the incoming airstream would be replaced within seconds by water vapour from the much more abundant liquid
water pool in the xylem on all sides of the borehole (Marshall et al. 2020). The sampled borehole lengths were thus sufficient
to reach equilibrium between the xylem water and the water vapour passing through the stems.

The isotopic composition of the equilibrated water vapour was determined in the field using a Picarro LI-2130i CRDAS water-
isotope analyzer plumbed into the manifold. We focus on $\delta^{18}O$ because it becomes less biased during  extractions than $\delta^{2}H$
(Barbeta *et al.*, 2018; Chen *et al.*, 2020). Every day the instrument was calibrated against two liquid working standards at three
different water concentrations using a Standards Delivery Module (SDM, Picarro). One standard was relatively enriched, with
$\delta^{18}O$ of -12.87 and $\delta^{2}H$ of  -94.28 ‰ . The second standard was relatively depleted.  Until May 7, 2018, the depleted standard
had $\delta^{18}O$ and $\delta^{2}H$ of -29.32‰ and -178.87‰, respectively. We then replaced the depleted standard by one that had $\delta^{18}O$ and
$\delta^{2}H$ of -31.05‰ and -205.64‰, respectively.

Isotopic composition of streamflow was measured from grab samples 224 times over this two-year period. The samples were
analyzed on a Picarro LI-2130i CRDAS water-isotope analyzer at the SLU stable isotope laboratory (Umeå, Sweden).
Measurements of isotopic composition of soil water were co-located with the xylem measurements, but began in 2018. These
measurements used equilibration devices (Soil gas lance BGLD 300, Meter Group, München, Germany) at 5 and 15 cm depths.
The 5-cm depth was replicated three times and 15-cm depth was represented by a single measurement. The soil samplers were
plumbed to ports on the same manifold as for the trees.


The isotopic composition of liquid water in xylem or soil water was inferred from the equilibrated vapour using the
temperature-dependent equilibrium fractionation factor (Majoube, 1971), as described for boreholes by Marshall et al. (2020).
One key difference, however, is that we did not measure borehole temperatures, which are necessary to parameterize the
equilibrium fractionation and thus to infer the isotopic composition of the liquid.  Instead, we calculated tree temperature based
on water vapor concentration, as measured by the Picarro analyzer. The inferred temperature was the one that would yield
100% humidity. This assumption of saturation in water vapor pressure is supported by measured and modeled water vapor
fluxes in boreholes of smaller trees at similar flow rates (Marshall et al., 2020).

Lastly, we calculated the seasonal origin index as:
Seasonal origin index (SOI) = $(\delta_x - \delta_{ann}) / (\delta_{summer} - \delta_{ann})$                    Eq. 1
if $(\delta_x - \delta_{ann})$ was greater than 0. If $(\delta_x - \delta_{ann})$ was less than 0, we used:



Seasonal origin index (SOI) = $(\delta_x - \delta_{ann}) / (\delta_{ann} - \delta_{winter})$            Eq. 2

where $\delta_x$ is the $\delta^{18}O$ of xylem water, $\delta_{ann}$ is the annual mean $\delta^{18}O$ of precipitation, $\delta_{summer}$ is the $\delta^{18}O$ of summer precipitation, and $\delta_{winter}$ is the $\delta^{18}O$ of winter precipitation. We estimated $\delta_{summer}$ and $\delta_{winter}$ by fitting a sine curve to the two years of precipitation data in Fig. 1B, yielding a $\delta_{summer}$ value of -8.88‰ and a $\delta_{winter}$ value of -17.02‰, Our xylem-water $\delta^{18}O$ values were always $> \delta_{ann}$ so we used Eq. 1, but the streamwater value was slightly less than $\delta_{ann}$, so we then used Eq. 2 (Allen *et al.*, 2019a).

**3 Results**

The summer of 2017 had regular rainfall throughout and served as a useful reference for the summer drought of 2018, during which rainfall was reduced to about half of normal (Fig. 1A). As expected, winter precipitation was more depleted in heavy isotopes than in summertime (Fig. 1B). Snowmelt flushed through the soils in May of both years, causing distinct spikes in the soil volumetric water content, especially at 50 cm depth (Fig. 1C). In the following months, water was lost from the soil by evaporation from the surface, lateral and downward drainage, and transpiration from all rooted depths. The lost water was partially replaced by summer rains. The slow decline in water content at 50 cm suggests that roots were taking up water from this depth during the dry summer of 2018, but perhaps not during 2017. The short-lived spikes in water content at 30 cm and 50 cm suggest that the biggest rain events were sufficient to induce a downward flux to these depths at least.



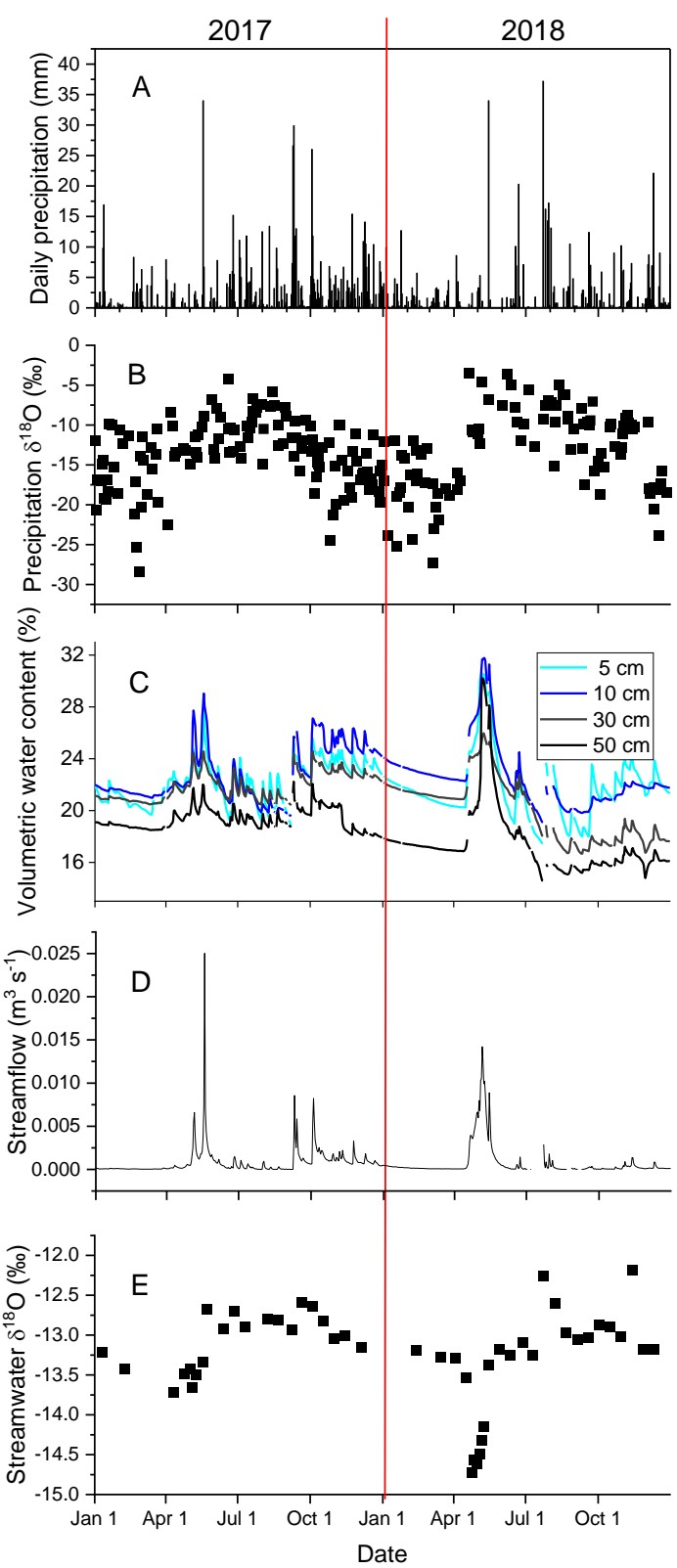





**Figure 1: Seasonal variation in environmental parameters during the normal (2017) and drought year (2018). A) Daily precipitation,**
**B) δ$^{18}$O of precipitation, C) Volumetric water content by soil depth, D) Streamflow rate, and E) δ$^{18}$O of streamflow. The vertical red**
**line divides 2017 (normal) from 2018 (drought).**

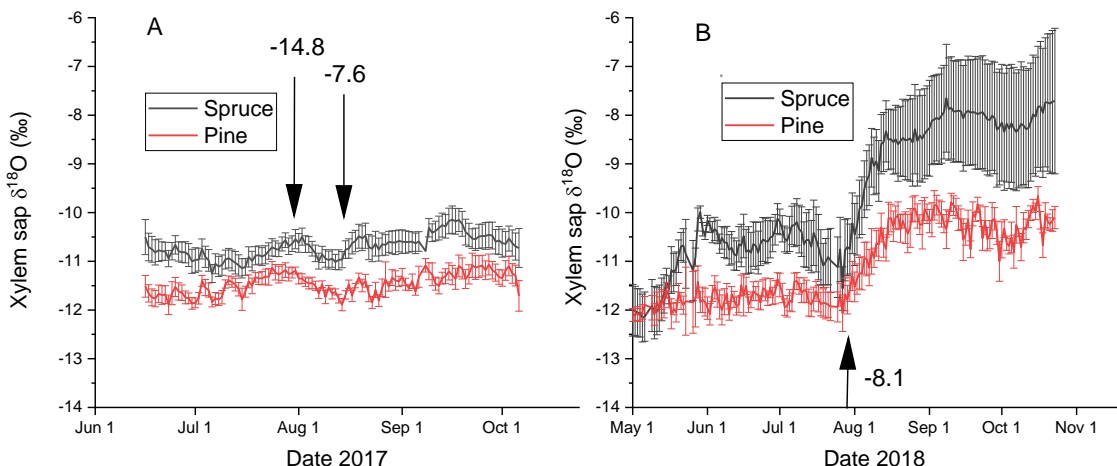

**Figure 2: A) Xylem sap δ$^{18}$O for spruce and pine during the average year of 2017. Means and standard errors were estimated from**
**five trees of each species. Two rainfall events with distinct isotopic signatures (‰) preceded isotopic shifts in the xylem sap. B) Xylem**
**sap δ$^{18}$O from same five trees of each species during the drought year of 2018. Pine and spruce clearly differ, but with strong seasonal**
**variation. The mean isotopic composition (‰) of the drought-breaking rains is shown.**

In 2017, the normal year, pine and spruce differed rather consistently in xylem-water δ$^{18}$O (Fig. 2A). Spruces (-10.00, SE
0.05‰) were always more enriched in $^{18}$O than pines (-11.27, SE 0.02‰) (Fig. 2A). However, the continuous data also detected
small seasonal differences in both species, especially following large precipitation events with distinct isotopic signatures. For
example, the two events marked by arrows in Fig. 2A show xylem-water responses to 12.5 mm of rainfall on 1 August (day
of the year, DOY 213) and 13.4 mm of rainfall on 10 August (DOY 222). The first rainfall, which was depleted in $^{18}$O (-
14.8‰), decreased the stemwater $^{8}$O values for several days afterward. The second rainfall, which was enriched (-7.6‰),
increased the stemwater $^{18}$O values for several days.
In the drought year (2018), the species difference in tree xylem-water δ$^{18}$O again appeared soon after snowmelt flushed the
soil profile (Fig. 2B). However, as the weeks passed without significant rainfall, the spruce δ$^{18}$O gradually fell into the pine
range, around -11.8‰, until their standard errors briefly overlapped late in July. This ended only when the drought was broken
(arrow, Fig. 2B), at which point both the pine and the spruce rose to more enriched values. The weighted-average isotopic



composition of the five drought-breaking rain events, which totaled 85 mm of precipitation, was -8.1‰, almost exactly the

isotopic composition of the spruce xylem water after the rains.  The pines increased by about half as much, to -10.2‰ (Fig. 2B).

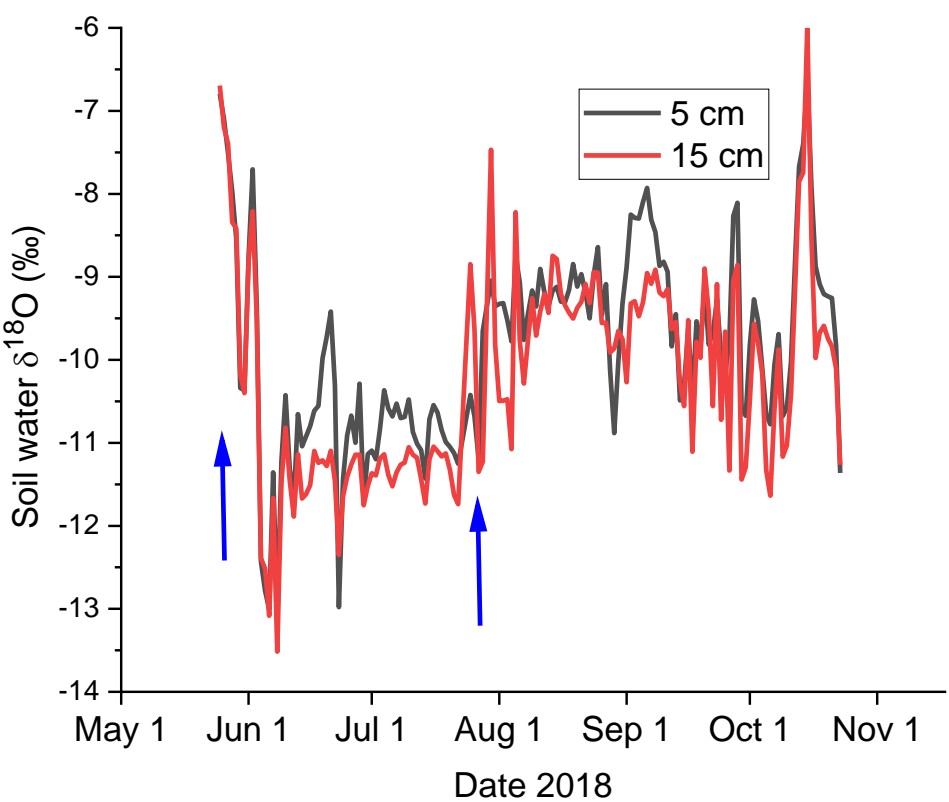


**Figure 3: Soil water $\delta^{18}O$ at 5 and 15 cm during 2018. Blue arrows show dates of the snowmelt flush and the rains that ended the drought. The soil water $\delta^{18}O$ tends to be higher at 5 than 15 cm, especially during the drought.**

In 2018, the 5- and 15-cm soil depths diverged in $\delta^{18}O$ after the snowmelt flush on 15 May (Fig. 3, arrow on left). During the

drought, the mean difference was +0.44 (SE=0.04)‰. The drought-breaking rainfall events (-8.1‰) were considerably more enriched in [18]O than the soil water pool just prior to the events (-11.5‰ average of two depths) and caused a distinct increase in the soil water values (Fig. 3, arrow on right).



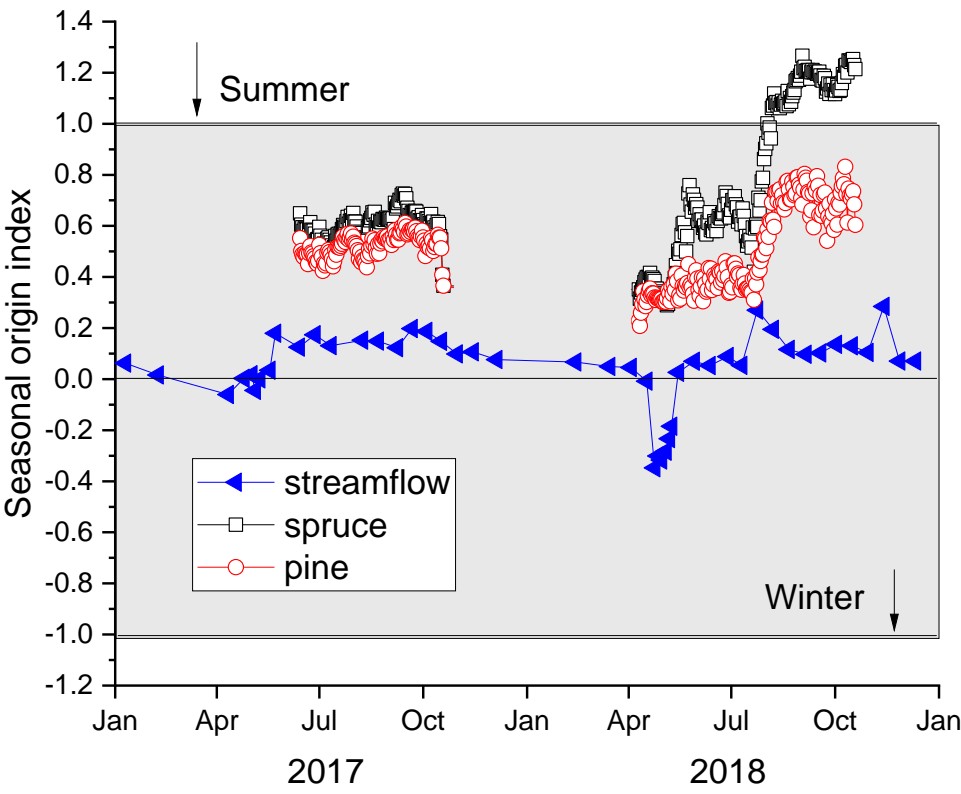

**Figure 4: Seasonal origin index of streamflow relative to water transpired by spruces and pines during the normal (2017) and drought year (2018). The index approaches +1 when δ¹⁸O represents summer precipitation and -1 when it represents winter precipitation. Transpiration is dominated by summer precipitation, especially in spruce trees after the drought, but streamflow represents an even mixture of seasonal sources for most of the year.**

For the xylem water used in transpiration, the seasonal origin index (Fig. 4) was consistently greater than 0, meaning that the xylem water was composed primarily of summer (SOI = +1) rather than winter precipitation (SOI = -1) throughout the study period. In 2017, the index consistently showed that spruce transpiration contained more water sourced from summer precipitation than did pine. In 2018, the drought year, the two species were even more different, however the spruce was again more cconsistent with summer precipitation than the pine. There was a slight shift toward greater sourcing of winter precipitation in the transpiration stream of both species at the very end of the drought. The SOI increased sharply in both species once the rains returned, exceeding +1 for the shallow-rooted spruce. In contrast, the streamwater was comprised of a nearly equal mixture of winter and summer precipitation for most of the year, with SOI averaging -0.02 (Fig. 4). Two exceptions were the period of snowmelt (April), especially in 2018, when streamwater approached the winter range, and the



period after the drought in 2018, when streamwater briefly approached the summer range. Thus, the trees transpired predominantly summer rain (positive SOI), especially the spruces, while streamflow was a rather even mixture of winter and summer precipitation.

## 4 Discussion

The "two water worlds" hypothesis has had great heuristic value for managers and the general public because it clarifies that transpiration and infiltration are not derived from a single well-mixed bucket (Penna *et al.*, 2018). Inside the research community it highlighted questions about fast and slow flowpaths through the soil, the use of different reservoirs to supply those flows, and the mixing of water pools during events (Penna *et al.*, 2018; Beyer *et al.*, 2020; Rothfuss *et al.*, 2020). This discussion actively continues. Our study confirmed the clear differentiation between transpiration and streamflow, but it also detected a clear distinction between transpiration of spruce vs. pine, as well as strong seasonal dynamics in transpiration water sources. In both tree species, transpiration favoured summer precipitation stored in the surface horizons, but spruce favoured it more. The observed seasonal differences followed large precipitation events, especially after the 2018 drought. The continuous, *in-situ* methods used here have provided a new level of temporal and species resolution for the description and modeling of these event-driven shifts, showing that the water worlds are diverse and dynamic.

The observed vertical profile in soil $\delta^{18}$O was consistent with the notion that a portion of summer precipitation moves vertically downward via translatory or "piston" flow, the slow process that replaces most or all of the water in soil pores as water is added from above (Fig 3). The surface layers held primarily water derived from the most recent rains (high SOI). Furthermore, the uppermost layer (5 cm) showed evidence of $^{18}$O enrichment due to evaporative processes later in summer. The minor effect of evaporation in the deeper soil layer (15 cm) resulted in more negative $\delta^{18}$O values (Laudon et al. 2004). Previous work at this site (Ameli *et al.*, 2021) sampled soils even deeper, from 70 and 90 cm, and found water more closely matching the annual mean precipitation (-12.5 to -13‰, Fig. 1B). In these deep soil layers, the SOI value ($\approx 0.16$) suggests a nearly even mixture of winter and summer precipitation, suggesting that some portion of the flow is not translatory and instead occurs rapidly via macropores. The presence of some macropore flow is supported by the event-driven upticks in volumetric water content even at 50 cm. This may help to explain the abundance of summer rainfall (high SOI) in the deep soil and streamflow samples. Note that the strong distinction between SOI of streamflow and transpiration in Fig. 4 tends to confirm the two-water worlds hypothesis, while the species difference calls for at least two separate transpiration sources.

Because the deeper soil layer was more negative in $\delta^{18}$O than the shallower layer (Fig. 3), we can also interpret isotopic composition of xylem water in terms of the depth of water sources. Specifically, we conclude that pines were drawing water from deeper in the soil profile than spruces at almost all times, reinforcing existing evidence of distinct rooting depths for these species (Bishop & Dambrine, 1995). The two exceptions were during the early spring, when the vertical profile was eliminated by the snowmelt flush, and during the peak of the 2018 drought, when the spruces appeared to shift briefly to deep-water uptake. Such a downward shift has previously been reported in Swiss beech trees, also during the drought of 2018. (Gessler *et al.*, 2021). Although the beeches shifted to deeper water sources, they were not able to maintain their transpiration rates. There



appears to be some ability to take up just enough water from by the deepest roots to survive drought conditions (Bachofen *et al.*, 2024). But it is worth noting that this "lifeline" will finally be broken if the drought becomes severe enough (Arend *et al.*, 2021, 2022).

There have been several earlier reports of species differences in xylem-water isotopic composition, generally based on instantaneous "grab" samples. These studies have often focused on water residence time, water age, or the seasonal origin index (Brinkmann *et al.*, 2018; Muñoz-Villers *et al.*, 2020), but see (Goldsmith *et al.*, 2011). In Switzerland, Allen et al. (2019) found underrepresentation of summer rainwater both in surface soils and in the xylem of the trees, especially in oaks and beeches. They attributed these results to macropore flow, which shunted summer rains rapidly through the surface soil rather

than storing them there. The co-occurring spruce trees contained more summer rainwater than other species, but not as much as reported here. Goldsmith et al. (2022) also found winter precipitation was overrepresented in nearly all trees, but the winter signal weakened as precipitation amount increased. Summers are obviously warmer and drier in Switzerland than in northern Sweden, but it is surprising that the amount of summer rainfall uptake was so different. SOI was also measured in streamflow in Switzerland (Allen *et al.*, 2019b), where summer and winter rains contributed approximately equally. The similarity in our

streamflow SOI despite a difference in our transpiration SOI is puzzling in light of mass-flow considerations, and calls for direct measurements of the missing evaporation fluxes, both from the canopy and from the soil.

    In summary, the two water world hypothesis was extended in two steps. First, we demonstrated that the transpiration flux varies between shallow-rooted spruce trees and deeper-rooted pines. Next we showed that these differences varied considerably over a severe drought, but also following rain events in an average year. Our continuous, in-situ sampling of xylem water

allowed us to move beyond the two water-world model to a more complex and dynamic view of hydrologic processes. These results can now be included in models of boreal forest transpiration and infiltration, but they also call for similar work at other sites, on other soils, and with different species mixtures.

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

**Acknowledgments:**



We recognize the contributions of Niles Hasselquist, who passed away before this manuscript was written. Niles led the field installation of this experiment and was responsible for many of the technical decisions that gave rise to this dataset.

The work was supported by the Knut and Alice Wallenberg Foundation (#2015.0047, 2018.0259, 2023.0245). We thank the SLU stable isotope laboratory (SSIL) staff for isotopic analysis and the staff contributing to the Swedish Integrated Carbon
Observation System (ICOS-Sweden) Research Infrastructure and the Swedish Infrastructure for Ecosystem Science (SITES), their financial support, their help in the field, instrument maintenance, and for providing data.

**Author contributions**

JM was responsible for the overall conceptualization. JM and HL acquired the funding. NK and JM analyzed the data. JM
wrote the original draft and all contributed to revision and editing.