# Peer review of "Beyond two water worlds: dynamic transpiration sourcing in a mixed-species boreal forest"

_EGUsphere, 2025_

## Author Comment (AC1)

**Reviewer 1:**

Marshall et al. present a two-year time series of stable isotopes of water in the xylem stream of two boreal forest tree species. The authors pursue three comparisons: 1) shallow vs. deep rooted species, 2) drought vs. non-drought year, and 3) water sources for streams vs. plants. It is this last point, written in terms of the "Two Water Worlds" hypothesis, that is the emphasis of the introduction and discussion.

**Synoptic Comments:**

This is largely a descriptive longitudinal study, with little formal statistical analysis. Not everything needs formal statistical analysis, but I believe that some opportunities to bring depth are missed. In doing so, one could provide additional insights into mechanism, which I think would be exciting. An example: It would appear that the authors have significant information on precipitation amount/stable isotopes of precipitation. How long does it take for the precipitation signal to propagate into soil, trees, or streamwater when there are distinct summer events?

To address the comment that our study was "largely descriptive," we have taken the reviewer up on their challenge, given as an example, to address the lags in passage of the the isotopic signals through the measured components of the hydrologic system. We began by tracing the drought-breaking event in 2018, which delivered a distinct isotopic signal against low background water concentrations. This yielded a new figure and abit of text describing flux dynamics, which we will include in the revised manuscript. Although we agree it would be interesting to analyze the entire two-year dataset, we have chosen not to do so because it would require a detailed analysis similar to Muesburger et al. (2020), but adding in isotopic data. And this would only address the soils, still not addressing dynamics in the xylem! In short, we think that an assessment of the 2018 drought-breaking pulse is within the scope of the current study, but are reluctant go further. The new analyses have given us opportunities to deal with several of the comments below. The data will be made available to the community and we are eager to support further analyses of this sort in the future.

There have now been many studies of "the two water worlds" hypothesis, which the authors are correct in couching as a heuristic means of proceeding. However, perhaps the turn away from this hypothesis (as noted by the authors themselves on L.56) is because it has reached its utility and there is a need for approaches that elucidate underlying mechanisms. In this sense, I was (personally) more excited by the \*continuous\* longitudinal nature of the study demonstrating what happens when there is drought and

relief from drought. I wonder if the authors might consider giving this more weight and reframe the introduction in those terms.

We are convinced that the two water worlds hypothesis remains a compelling starting point for this analysis, particularly for readers who are less familiar with the literature in this field, if only because it raises questions about how many water pools are necessary, how they might be measured, and how dynamic they are. We agree that we did not adequately address these broader questions in the introduction and discussion, and will do so in the revision.

Precipitation, soils, trees and streams are never all shown on the same plot; this seems like an opportunity missed, given how few studies have been able to collect all of these simultaneously!

Interesting point, thanks for highlighting this opportunity! We will separate the non-isotopic data into a new Figure 1 and present all of the isotopic data in a new Figure 2, including the xylem and soil water data that are currently in Figs. 2 and 3. We will further present these isotopic data in figure describing the isotopic excursions after the drought breaking rains in 2018.

A primary comparison of this paper is 2017 vs. 2018, which are each given their own panels. An alternative approach would be to have a single panel, with data colored by year to facilitate easier comparison. In addition (and this is would be very useful), recommend that authors shade with transparency the time period of interest (summer) to make interpretation of results easier.

We tried combining the two years on one plot, but found the error bars difficult to see. We thought the error bars were important, especially when the spruce starts to fall toward the pine at the end of the 2018 drought. We will emphasize the error bars more and emphasize the matching y-axes between the panels. We agree to shade the summer period and emphasize that the data series began on different dates.

**Introduction**

L. 30. I recommend that you modify the first sentence, which neglects to mention the rest of the planet's fauna. Perhaps more importantly, the study is not actually about competing uses of water by vegetation and humans.

**Agreed. Will fix.**

L. 43. I recommend that you consider that the citation of Brooks et al., while true, neglects the primary proposed hypothesis, which was a temporal offset in the refilling of empty soil water storage (winter) from the timing of its subsequent use.

**Will add this mechanism to the description.**

L. 44. Similar to the comment above about Brooks et al., the description of Allen et al. is correct, but perhaps an incomplete interpretation. In fact, it would appear that soils hold different amounts of summer precipitation and in drier sites, tend to hold more winter precipitation (Allen et al. in HESS). This (and proposed climatological mechanisms) are proposed in Goldsmith et al. (2022 in GRL) and Floriancic et al. (2025 in Ecohydrology).

**Will add a statement to this effect.**

L. 58. I recommend that you consider whether recent studies have not discussed water sources in terms of "water worlds" because the framework may not have utility in advancing our understanding of the processes that underly the observations.

We were perhaps overly careful. Will now strengthen these statements in the revision

**Methods**

L. 80. Recommend that you provide the difference in (e.g.) summer rainfall between 2017 and 2018 as a total and as percentage of the annual. Otherwise, it is hard to contextualize the severity of the drought from Figure 1 alone, wherein there does seem to be episodic rainfall.

We agree that Fig. 1 is not very compelling on its own. There are now several papers describing the drought at this site and will use them to inspire a better way to provide context.

Figure 1. Recommend you add more dates to x-axis, particularly top panel, where it would be nice to see times more closely. Additionally, consider merging 2017 and 2018 into one panel with different colors, as that is a primary comparison.

**Agreed.**

Figure 1. Recommend reconsidering the red line. More useful than delineating a calendar year would be to delineate summer months through gray shading.

**Agreed.**

L. 84. Recommend you delineate how long sensors were in place before measurements began.

**Agreed.**

L. 105. Recommend that you specify that all isotopes are provided per mille relative to V-SMOW.

**Agreed.**

L. 109. Recommend that you offer additional details on sample handling for water samples. I assume sealed in glass/plastic vials and stored in a cool setting until analysis.

**Agreed.**

L.111. The study is actually all the stronger for at least having some measure of the soil water. It's striking that it takes halfway through the methods to read this – it's a much more complete picture with this in place and it should be mentioned in the introduction.

**Agreed.**

L. 113. Recommend that you compare how the depths of the soil water sampling match with what is known about the depths of soil water use by the two contrasting species.

**Agreed.**

L. 116. How does the isotopic value of water vapor in soil compare to what we would expect be available to plants. Or, in other terms, water that is more or less mobile? Recommend that the authors comment.

This question has been answered by others who use soil equilibration methods. Will explain and cite.

L. 125. Since SOI is a comparison with precipitation and xylem water reflects a potentially evaporated source water signal (soil), many studies compensate for this evaporation (e.g., see original work by Allen et al. 2019 in HESS). Recommend pursuing this approach or at least confirming that it does not change your interpretation, especially in 2018 when you would expect drought to have an impact.

We agree that we should mention this important point. We will explain that the approach to the precipitation isotope value in the spruce xylem water after the rains in 2018 suggests that at least in that event, there was not much evaporation. Will also cite Allen et al.'s observation that the canopy interception does not have much effect. This could be quite different if the events were small and infrequent.

L. 130. Recommend that you include details on how the precipitation isotope data were collected. What device? Are the isotope ratios amount weighted? How often were samples collected? How were they stored?

Will provide these details.

**Results and Discussion**

L. 136. Recommend quantifying precipitation amounts and providing comparisons.

L. 139. While this is almost certainly true, it's so true as to be obvious. On the other hand, is physical surface evaporation a consideration in this ecosystem? Recommend revising this sentence.

L. 142. It should be relatively easy to calculate a minimum event size needed to percolate to the different sensor depths. Recommend that the authors consider as much in order to bring depth of understanding to this analysis.

This is actually a bit tricky with the canopy interception and the water absorption in the forest floor, however it has been done before, and near the current study site. We will generate an estimate and compare it to the time delay in the soil isotope shift, while noting that the soil may lag because of limited hydraulic conductivity.

L. 153-154. This number "(-10.00, SE 0.05‰)" and similar numbers are hard to interpret. Recommend specifying if it's an annual mean and using plus/minus per typical convention, then specifying that it is standard error.

**Will do.**

L. Figure 2. I've always been a little uneasy with the idea of calling it sap and in fact, this is the only place where it is referred to as such. Recommend "Xylem water" or "Xylem water vapor" for consistency with paper. Recommend indicating per mille after noting isotope ratios of particular events, for sake of clarity.

**Agreed.**

L. Figure 3. Are these continuous measurements? If not (or if average in some way), recommend adding points on the lines to clarify sampling interval.

**Will do. They are daily averages.**

L. 184. Why refer to it as "the xylem water used in transpiration" here and not elsewhere? This confused me; recommend you clarify if possible.

Can instead emphasize that xylem water goes to transpiration in the introduction and delete reference to transpiration here..

L. 188. To me, it would appear that the trees had lower SOI values at the start of the growing season in 2018 in general, as well as a change given drought. Recommend you consider commenting.

**Can emphasize here that the data began much earlier in the year in 2018 than 2017.**

L. Figure 4. Here the seasonal origin index is >1 in late 2018, indicating that the xylem water is in excess of the summerP isotope value. This would be an argument for providing more information on the calculation of the precipitation isotope sine curve.

**Agreed.**

L. Figure 4. What is the SOI of soil water? Recommend adding this to the figure.

**Agreed.**

L. 214. More probably, an SOI near zero is an almost infinite possible mix of spring, summer, fall and winter waters.

**Right. Will modify.**

**Relevant recent literature:**

Floriancic et al. (2024) Isotopic evidence for seasonal water sources in tree xylem and forest soils

Kinzinger et al. (2025) Continuous In-Situ Water Stable Isotopes Reveal Rapid Changes in Root Water Uptake by Fagus sylvatica During Severe Drought

Brighenti et al. (2024) Snowmelt and subsurface heterogeneity control tree water sources in a subalpine forest

Sprenger et al. (2025) Opportunistic short-term water uptake dynamics by subalpine trees observed via in situ water isotope measurements

Thanks for these excellent suggestions! We will certainly cite some, at least.

---

## Author Comment (AC2)

**Reviewer 2:**

This manuscript presents an excellent dataset: two years of continuous isotope measurements in precipitation, soils, trees, and stream water in a boreal mixed forest. The contrasts between spruce and pine, and between a normal (2017) and extreme drought year (2018), are highly valuable. The results go beyond the traditional "two water worlds" framing by showing that transpiration is species-specific and dynamic, shifting with drought and rewetting.

**General comments**

This is a strong and timely paper. The continuous isotope record across two contrasting years is a major contribution. The paper would be significantly strengthened if the authors (i) reframe the introduction toward dynamics, (ii) add quantitative analyses (event lags, rainfall thresholds, precipitation totals, unmixing), (iii) clarify methods (ideally with a schematic of the sampling design and photos of the site), and (iv) improve figure compilation.

We will further emphasize dynamics. The photo is a good idea. We will add a Gantt chart to the supplement showing when each set of measurements were made.

I also wonder whether the Seasonal Origin Index (SOI) is the most informative metric. The precipitation isotopic data do not follow a clear sine-shaped seasonal pattern, which may limit interpretability. A summer–winter endmember unmixing approach applied per time step could better capture event-driven dynamics, highlight species contrasts, and allow uncertainty estimates (e.g., via Bayesian or bootstrapped mixing models). The SOI metric could still be retained for comparability.

Summer/winter unmixing is certainly possible, but it would not add so much to the SOI analysis, which essentially does the same thing. We will provide clearer references of the unmixing approach as an alternative in the discussion. We agree that it will help to present the sine curve fit to show where the "non-sine" behavior occurs and how it influences the results. In addition, we have added a description of the dynamics during the drought-breaking event in 2018.

The continuous monitoring through drought and recovery is the most exciting aspect of the study. While the "two water worlds" hypothesis provides a useful background, it may have reached its heuristic limit. The dataset is better framed in terms of temporal dynamics and species contrasts, as already suggested by Reviewer 1.

We will place a heavier emphasis on the dynamic responses.

I encourage the authors to be cautious with terms such as "water pools," which can imply complete separation of sources. Water in soils and catchments mixes continuously across depths and time. It is not black and white (not purely winter vs. summer or deep vs. shallow) but a gradient of interactions and mixtures (e.g., Dubbert et al., 2019). Acknowledging this explicitly and noting that such behavior can be simulated with mechanistic models would provide a more nuanced interpretation. For instance, Meusburger et al. (2022) showed with modeled root water uptake that both shallow- and deep-rooted species switched to deep soil water during drought but returned to recent shallow water after rewetting. An extension of this model with isotope transport (LWFBrook90.jl) can reproduce the distinct signatures between species and residual (blue) water. This suggests that physically based models can capture these dynamics without invoking a rigid "two water worlds" separation, reinforcing a process-based interpretation of water sourcing.

The Meusburger ms. provides a good example of a non-isotopic, soil-focused analysis of water consumption by forests from a broad range of soils. It would be interesting to try analyzing these data in this way. But it seems beyond the scope of the current manuscript, especially considering that we wish to emphasize the xylem data. We will provide the documented data and contact potential collaborators to see if they would be interested in working on such an analysis for another paper.

The manuscript is largely descriptive. While not a weakness, adding straightforward analyses would strengthen mechanistic insight. I also second the comments of Reviewer 2 regarding more quantitative evaluation. Because the soil isotope equilibration devices were only installed at 5 and 15 cm depth, please discuss how you can be confident that a meaningful isotopic gradient existed along the full profile, particularly given that spruce and pine likely access water from different depths. Providing basic information on total soil depth would also be helpful.

We have added a brief description of the pulse after the 2018 drought was broken, which leads into a more mechanistic set of interpretations. The soil data were so limited that we could not support much mechanistic interpretation of them. We have clarified this limitation in the text. Soil depth data will be taken from earlier hydrologic studies at this site, which show exponentially declining hydraulic conductivity with depth (Bishop et al. 2004), which results in isotopic mixing primarily in the upper layers (Laudon et al. 2004).

The observed vertical profile in soil  $\delta^{18}$ O is consistent with piston flow (line 208), but a simple quantitative check could make this conclusion more robust. For example, compare pre- and post-event soil  $\delta^{18}$ O values and volumetric water contents to estimate the fraction

of soil water replaced during major rainfall events, providing quantitative support for Figure 3.

We will add a brief description of this phenomenon to the new text focusing on the breakthrough after the rains in 2018. We will need to use forest floor data from studies at adjacent sites and the time to breakthrough at the 5-cm soil depth. Particularly interesting in this new analysis is the  $\delta^{18}$ O of the 5-cm soil probes, which did not rise as high as the heavy rains, suggesting that only half (53%) of the water was replaced after the rains. This point will be added.

At line 215, the manuscript states that the SOI value (~0.16) in deep soil layers indicates a nearly even mixture of winter and summer precipitation and interprets this as evidence for macropore flow. This pattern may not exclusively indicate macropore flow; it could also reflect seasonal removal of isotopically heavier summer water by transpiration, leaving lighter winter-like water behind. Please discuss both processes as potential contributors.

**Agreed. Will do.**

Regarding Fig. 4, the strong distinction between SOI of streamflow and transpiration is interesting but should not be overinterpreted as evidence for fully distinct water worlds. A continuum shaped by root distribution, water availability in soil layers, and temporal dynamics of inputs and outputs is a more realistic interpretation.

This is the point we were trying to make, apparently not clearly enough. Will strengthen it

**Additional small comments:**

- Consider combining 2017 and 2018 into single panels.
  - As in R1 response, it became too busy.
- Consider adding all compartments (precipitation, soils, trees, streams) into one composite figure (Reviewer 1 suggestion). Do the mixtures lie within the sources?
  - o Will do.
- Line 225: correct to Gessler et al., 2022.
  - Yes, will fix
- Clarify in Methods how precipitation isotope sampling was done (collector type, frequency, storage) to ensure reproducibility.
  - o Will do.

- Report isotope analyzer precision (e.g., ±0.1 %) and sample handling details (sealed vials, storage conditions) so readers can judge data quality.
  - o Will do.
- In Figure 3, if soil data are discrete, mark sampling points; this will make profiles easier to interpret.
  - o Yes, these are daily means. Will present the points.
- Add seasonal shading or vertical bars to Figure 1 for summer months to highlight the drought period visually.
  - o Agreed.
- Provide average temporal resolution of stream water sampling (e.g., weekly/biweekly).
  - o OK, and will note that the stream ran dry during the 2018 drought.

Bishop, K., Seibert, J., Koher, S., and Laudon, H.: Resolving the Double Paradox of rapidly mobilized old water with highly variable responses in runoff chemistry, Hydrol Process, 18, 185-189, 10.1002/hyp.5209, 2004.

Laudon, H., Seibert, J., Kohler, S., and Bishop, K.: Hydrological flow paths during snowmelt: Congruence between hydrometric measurements and oxygen 18 in meltwater, soil water, and runoff, Water Resour Res, 40, 10.1029/2003wr002455, 2004.

---

## Author Comment (AC3)

**Reviewer 3:**

The manuscript "Beyond two water worlds: dynamic transpiration sourcing in a mixedspecies boreal forest" presents a unique dataset of in situ measured xylem water isotopic signatures for two boreal tree species spanning two contrasting growing seasons.

The manuscript is well written, clearly structured and pleasently concise. Overall, I would recommend this paper for publication after some minor revisions.

My main point of critique for the current manuscript would be the incidental treatment of the soil water isotopic signatures. Clearly, the isotopic signatures of soil water are of paramount importance to explain the observed xylem water isotopic signatures, yet, the authors did not report any soil isotopic signatures for the first year and only measurements from 5 and 15 cm soil depth for the second growing season. It is safe to assume, that both tree species' root systems are likely to exceed these depths.

We agree that more soil depths and from both growing seasons would have been valuable. We did not treat them as of paramount importance because we originally focused the study on comparing the precipitation to the tree stem and stream water. As noted above, we have now added a section describing in more detail the passage of the rainfall pulse through the system after the drought was broken in 2018. In that analysis, we emphasize that the soil isotopic data approached, but did not reach the values from the precipitation and xylem water. Evaporative modification along the path from precipitation to the upper soil horizons would more likely have enriched the isotopic signal, leading to higher values in the soil rather than lower. The soil data are more consistent with the notion that only about half the soil water was replaced by piston flow (see R2 above). We will present these issues in a new paragraph in the discussion. The new analysis integrates the soil data more effectively than before, but recognizes that there are limits to what we can do with the data in hand.

Furthermore, I am a bit sceptical about the use of BGDL 300 soil gas lances for the measurement of soil water isotopic signatures. Are there any previous publications that have used and validated these devices for the measurements of stable water isotopes? If yes, I would like to see a reference for that, if no, the manuscript should include some more details on the setup such as the material and dimensions of the BGDL 300. Did you also use heating cable for these lines? Did you do any comparisons to more estblished ways of measuring stable water isotopes of soil water?

This is true. We know of no previous publications using this device. However there are multiple publications describing similar soil equilibration methods. In any case, we agree that more detail is required and will provide it in the revision.

Looking at the results in Figs. 2 and 3, I see that after the drought ending precipitation event with -8.1%  $\delta$ 18O, which shifted the xylem isotopic signature of spruce xylem water towards values around -8%  $\delta$ 18O, somehow both of your observed soil depths did not exceed -9%  $\delta$ 18O. This would imply that spruce had to source nearly all of its water from above 5 cm of soil depths and that even 85mm of rain were not enough to flush the soil water isotopic signatures to a depth below 5 cm. Or could it be that your measurements of soil water isotopic signatures are somewhat biased?

Yes, they could be biased. In the additional text describing the breaking of the drought, we note that the probes gave unsteady results during and immediately after the heavy rains, but appeared to stabilize after this time. We speculate that liquid water may have flowed down along the probe surface. This question about piston flow in the presence of a litter layer and canopy interception has come in all the reviews. We will add a rough analysis of it, recognizing that the soil data are not limited. We emphasize again that the primary focus on the manuscript was on the xylem water and streamflow, as noted just below.

I am fully aware that the focus of this study was on xylem water isotopic signatures and the results indicate an impressive successful long term application of the borehole technique for natural abundances of 18O. However, I think the soil water isotopic signatures within this study deserve and require a more detailed discussion. Do you trust the results? Can you recommend the use of BGDL 300 soil gas lances for soil isotopic measurements?

R2 made some similar points. We will add more description of the soil results and of the device.

Apart from that, I have the following minor comments:

line 92: Please also specify the inner diameter and wall thickness of your PFA lines.

OK, we will add this info.

line 95: If possible, please specify the type and manufacturer of the solenoid valves.

**OK, we will add this info.**

lines 92-103: The description of the setup is a bit unclear to me: did you just draw atmospheric air from one end of the borehole through the tree and into the analyzer, or was there an additional dry air supply connected to the entry side of the borehole (as. depicted in Fig.2 of Marshall et al. (2020))?

We will clarify in the text that we simply drew ambient air into the borehole trusting that it would equilibrate with the xylem water as modeled and empirically demonstrated in Marshall et al. (2020).

lines 103: You say you "focus on  $\delta$ 18O because it becomes less biased during extractions than  $\delta$ 2H", but these effects are likely to be limited to cryogenic extractions, which are not part of your study. Volkmann et al. (2016) reported a missmatch between in situ  $\delta$ 18O meausrements with a CRDS compared to IRMS meausrements of destructive samples, while  $\delta$ 2H showed no such missmatch. Kinzinger et al.(2024,

https://doi.org/10.1093/treephys/tpad144), using the same in situ methodology as Volkmann et al. (2016), focused their analyses to  $\delta 2H$  because  $\delta 18O$  showed a lower accuracy and higher drift. It would be interesting to know how the results for the two isotopes compare in your study. I would love to see a dual isotope plot of your in situ measurements (maybe as a supplement figure), just to get an idea on the cababilities of the borehole technique.

The potential problems with isotopic analysis occur not only with cryogenic distillation, but also with any sample that contains sufficient organic interferents. We have added a statement to this effect, speculating that this was what happened to our  $\delta^2H$  data. We are aware that Herbstritt et al. (2024) observed no such interference in on-line measurements of garden vegetables and deciduous trees, but a conifer tree may be quite another thing. In any case, we have added dual isotope plots into the supplementary materials to show that the precipitation data look fine but the borehole data have their intercept shifted, consistent with the organic interference hypothesis.

Fig.3: As you mentioned that there were three repetitions for the depth of 5 cm - what is the grey line showing? The mean of all three repetitions? Could you also indicate the range of the three repetitions, or just show each of the measured time series with a seperate line?

Yes, the grey line shows the mean of the three probes. Examination of the individual time series showed that one of the probes was the source of much of the noise. The noise occurs primarily on days with heavy rains, consistent with the idea that the probes might provide a preferred pathway for infiltration if installed incorrectly. We have dropped this noisy probe from the main analysis. However, we present all of the probes, including the noisy one, in a figure in the Supplementary Materials.

line 191: Is -0.02 actually the average streamwater SOI? In Fig. 4 it looks like the streamwater SOI is positive most of the time - or did you use a mass weighted average?

**Yes, we used a weighted average and we now say so.**

line 199: In my opinion, the dichotomy between "transpiration" and "infiltration" seems misplaced, since transpiration is also likely to be sourced from water that infiltrated into the soil. Maybe "(deep) percolation" or "ground water reacharge" would be better terms than "infiltration".

**Agreed.**

lines 245-247: Speaking of models: Could you provide (or at least archive) the data presented in Figs.1-3 as well as additional site information such as a more detailed description of the soil profile and stand properties (basal area, leaf area index, any known information on root density distributions) and (if by any chance available) sapflow or dendrometer data in such a manner, that future modellers could use them in a quantitative way? Currently, the manuscript is missing a data availability statement.

Yes. We hope that these new data will be used by modellers and others in future work. Therefore we will provide and archive them in an open access depository. Previous work at this site has provided considerable data on soil and stand descriptions, which we will summarize in a table in the supplementary materials. We will also provide a data availability statement.

**Herbstritt B, Wengeler L, Orlowski N. 2024**. Coping with spectral interferences when measuring water stable isotopes of vegetables. *Rapid Communications in Mass Spectrometry* **38**: e9907.